# Effect of an Antioxidant Supplement Combination on Boar Sperm

**DOI:** 10.3390/ani12101301

**Published:** 2022-05-18

**Authors:** Ivan Galić, Saša Dragin, Ivan Stančić, Milan Maletić, Jelena Apić, Nebojša Kladar, Jovan Spasojević, Jovana Grba, Zorana Kovačević

**Affiliations:** 1Department of Veterinary Medicine, Faculty of Agriculture, University of Novi Sad, 21000 Novi Sad, Serbia; ivan.galic@polj.uns.ac.rs (I.G.); dr.ivan.stancic@gmail.com (I.S.); spasojevic.jovan@polj.uns.ac.rs (J.S.); zorana.kovacevic@polj.edu.rs (Z.K.); 2Department for Animal Sciences, Faculty of Agriculture, University of Novi Sad, 21000 Novi Sad, Serbia; jovana.grba@stocarstvo.edu.rs; 3Faculty of Veterinary Medicine, University of Belgrade, 11000 Belgrade, Serbia; maletic@vet.bg.ac.rs; 4Scientific Veterinary Institute Novi Sad, 21000 Novi Sad, Serbia; laskov2@neobee.net; 5Department of Pharmacy, Faculty of Medicine, University of Novi Sad, 21000 Novi Sad, Serbia; nebojsa.kladar@mf.uns.ac.rs

**Keywords:** pig, boar fertility, antioxidant status, sperm DNA status, sperm motility

## Abstract

**Simple Summary:**

Boar spermatozoa can be exposed to many harmful factors. The low antioxidant capacity of spermatozoa makes them susceptible to reactive oxygen species. The protection of spermatozoa from these harmful effects lies in the seminal plasma, which is enriched by antioxidant enzymes. This study aimed to measure the antioxidant status of boar seminal plasma, and to determine spermatozoa DNA status, and the total number of motile spermatozoa after the concurrent joint application in the boars’ diet of two commercial products with antioxidative potential.

**Abstract:**

The study was conducted on a commercial pig farm located in Serbia. Thirty Duroc or Landrace breed boars were randomly selected for this study. The experimental group was fed a compound feed with added organic selenium and Oxynat 3D. The antioxidant status parameters of boar seminal plasma were evaluated using a biochemical analyzer and commercial Randox kits. The sperm chromatin structure assay (SCSA) using flow cytometry (FC) provided information about spermatozoa’s DNA status. Additionally, the total number of motile spermatozoa and spermatozoa kinematic parameters were measured using the computer-assisted sperm analysis (CASA) system. The aim of this study was to improve the parameters of semen by combining two preparations that have a potential antioxidant effect, but also to establish the level of various antioxidant enzymes in native sperm. There was no statistically significant difference in total antioxidant capacity and glutathione peroxidase activity in the seminal plasma obtained from the experimental and control groups of boars. Regarding the superoxide dismutase activity, the research results showed a difference in the control group compared to the experimental one. Moreover, spermatozoa DNA fragmentation and the total number of motile spermatozoa showed statistically significant lower and higher values, respectively, in experimental compared to the control groups. The combination of these two preparations shows significantly enhanced vital parameters of semen. To the best of our knowledge, this study is the first in which the ejaculate parameters were examined after the application of a combination of these two antioxidant supplements.

## 1. Introduction

The main reasons for using artificial insemination (AI) in pigs are to increase reproductive efficiency and rates of genetic improvement. Obtaining good reproductive material from boars includes continuous improvements and adopting new procedures and technologies in the assessment of good quality semen for AI [1,2,3]. It is well known that antioxidant supplementation in humans and animals has a lot of benefits, but the potential clinical benefit deriving from these supplements is still under wide debate, due to the need to find optimal levels of antioxidants that will improve body functions and even the male reproductive system [4,5,6,7].

Oxidative stress occurs when the production of reactive oxygen species (ROS) overcomes the internal antioxidant defenses of the biological system and, thus, leads to cell damage and death [4]. Low concentrations of ROS play an important role in physiological processes such as capacitation, acrosome reaction and fertilization [5]. In boar spermatozoa, a moderate change in intracellular ROS levels modulates the activity of protein kinases and phosphatases that are involved in spermatozoa capacitation, while partial disruption of disulfide bonds in spermatozoa proteins and an increase in intracellular ROS levels could be involved in the decondensation of spermatozoa pronuclei after fertilization [6]. ROS attack results in decreased spermatozoa motility, axonemal damage, decreased spermatozoa viability, and increased mid-piece spermatozoa morphological defects with deleterious effects on spermatozoa capacitation and acrosome reaction [7]. The addition of antioxidants to boar feed protects spermatozoa from oxidative damage and prolongs their lifespan [8]. Various dietary supplements have shown good effects on the antioxidant status of boars, such as the addition of fish oil [9], vitamin E [10,11], organic selenium [12], and wild ginseng root extracts [13].

Antioxidant enzymes have an important function in protecting the spermatozoa from ROS that accumulate in the seminal plasma. The most effective antioxidant enzymes include superoxide dismutase, catalase, and glutathione peroxidase, among which superoxide dismutase is considered the most important antioxidant [14,15]. Superoxide dismutase activity is increased with elevated oxygen concentration in vivo, and its biological function is manifested through the transformation from superoxide radicals (O_2_-) to less toxic hydrogen peroxide (H_2_O_2_) [16]. Glutathione peroxidase becomes associated with the sperm plasma membrane and accompanies spermatozoa in their transit, protecting the cells from the harmful effects of peroxides [17]. The seminal plasma contains many chemical components with antioxidant properties, enzymatic and non-enzymatic [18], both of which make up its total antioxidant capacity.

The sperm chromatin structure assay (SCSA) using flow cytometry can produce information about sperm DNA status and provide meaningful biological information on sperm nuclear DNA defects [19]. Alongside computer-assisted sperm analysis (CASA), which allows more detailed and precise sperm motility assessment, flow cytometry is considered an important tool for determining the fertilizing capacity of spermatozoa [20]. Boar semen with a high percentage of DNA fragmentation produces low farrowing rates [21] and small litter size [22]. Meanwhile, a human study on the combination of antioxidants for two months, and another similar study using a cocktail of various antioxidants for three months, resulted in enrichment and improvement of sperm DNA status in cases unexplained infertility [23,24]. On the other hand, it appears there are no substantial reports demonstrating the benefit of antioxidant supplementation on boar spermatozoa DNA integrity in vivo.

In line with these facts, the current study aims to determine the antioxidant enzyme activity (superoxide dismutase and glutathione peroxidase), and the total antioxidant capacity in boar seminal plasma, and the sperm DNA status, the total number of motile spermatozoa and spermatozoa kinematics parameters after boars consumed feed with two supplements, organic selenium and the plant extract-based additive Oxynat 3D. The research hypothesis is that the experimental group will have higher semen quality with regard to total motility, DNA status and antioxidant protection of seminal plasma.

## 2. Materials and Methods

### 2.1. Animals, Semen Collection and Preparation

Boars (*n* = 30) were stabled in individual pens of 8 m^2^ area with a concrete floor. The stable was equipped with an automatic microclimate control. The average age of boars was 22 months. Drinking water was available ad libitum. The animals were subjected to a routine disease prevention program and regular veterinary care. The experimental group of animals consisted of 15 boars of either Duroc or Landrace breeds that were fed a compound feed containing 250 g/t of organic selenium (LFA Lesaffre, Toluca, Mexico) and 500 g/t of Oxynat 3D (Phytosynthese, Mozac, France). The control group consisted of 15 Duroc or Landrace boars that were fed the same compound feed but without the additives. Oxynat 3D is composed of four plant extracts from grape seeds, citrus, marigold, and rosemary. The levels of action for Oxynat 3D are: (1) capturing free radicals (initiation phase); (2) propagation (limiting the propagation speed); (3) detoxification (lowering the content of toxic compounds).

Semen samples (Figure 1) were collected on the commercial pig farm in the autumn of 2020. Those native sperm samples were taken on the first day of the study (1st September), and then every fifteen days for two months (a total of five samplings), while fresh diluted boar semen was controlled four times within an eight-week interval (the date the study started, and afterwards on every twentieth day). Semen was collected by manual fixation of the penis in a plastic bottle for boar semen (Minitube, Tiefenbach, Germany). Upon ejaculation, each semen sample was diluted 1:2 in a commercial extender (Vitasem, Magapor S.L., Zaragoza, Spain), and transported to the laboratory. The dynamics of collecting samples for testing was harmonized with the program of regular collection of semen from boars for the needs of the AI reproduction center, so that boars would not be used additionally.

Native sperm samples were transported to the Laboratory for Animal Reproduction at the Department of Veterinary Medicine (Faculty of Agriculture, Novi Sad). The seminal plasma was separated from the spermatozoa by double centrifugation of the boar semen at 2000× *g* for 10 min at room temperature, after which approximately 4 mL of seminal plasma were transferred to clean test tubes and then stored at −20 °C until analysis. Seminal plasma was used to test for the superoxide dismutase activity, glutathione peroxidase activity and total antioxidant capacity.

The samples of fresh diluted boar semen (the final concentration was 35 × 10^6^ spermatozoa per mL) were transported to the Laboratory of Reproduction at the Scientific Veterinary Institute Novi Sad. The samples of fresh diluted semen were used to determine sperm DNA status and the total number of motile spermatozoa.

The samples were transported in portable refrigerating equipment, Klimabox (Minitube, Tiefenbach, Germany), within 60 min from the moment of sample collection. During transport, samples were kept at 16–18 °C. Semen quality analysis was performed on arrival.

### 2.2. Determination of the Antioxidative System in Seminal Plasma

The antioxidative system in seminal plasma was monitored by determining the superoxide dismutase, glutathione peroxidase, and total antioxidant capacities. The superoxide dismutase activity, glutathione peroxidase activity, and total antioxidant capacity of the seminal plasma samples were each determined spectrophotometrically on an Olympus AU 400 automatic biochemistry analyzer (Olympus, Tokyo, Japan) using commercial Randox ^®^ kits (Randox Laboratories Ltd., London, UK). Namely, the superoxide dismutase activity was determined by a commercial Ransod ^®^ kit (Cat. No. SD124; Ransod Control reference material SD126, Randox Laboratories Ltd., London, UK), the activity of glutathione peroxidase by a commercial Ransel ^®^ kit (Cat. No. RS504; Ransel Control reference material SC692, Randox Laboratories Ltd., London, UK), and the total antioxidant capacity concentration was determined with a commercial Total Antioxidant Status ^®^ kit (Cat. No. NX 2332; Randox Total Antioxidant Control Cat. No. NX 2331, Randox Laboratories Ltd., London, UK). Superoxide dismutase activity was expressed in U/mL, glutathione peroxidase activity in IU/L, and the concentration of total antioxidant capacity in mmoL/L of seminal plasma.

### 2.3. Spermatozoa Motility Determination

Immediately after arrival in the laboratory, the samples of fresh diluted semen were reactivated in a water bath at 35 °C for 30 min before the testing began. The total number of motile spermatozoa and assessment of kinematic variables of spermatozoa were measured using the CASA system with an integrated software system for sperm analysis (ISAS, Projser, Madrid, Spain). Determination of spermatozoa motility was performed on four-chambered microscopic ISAS disposable slides (ISAS D4C20, Projser, Madrid, Spain) with 20 µm chamber depth and a volume of 5 µL, which was marked by 7 visual fields provided for recording. Spermatozoa kinematic parameters (VCL—curvilinear velocity, VSL—straight-line velocity and VAP—average path velocity) are expressed in µm/s.

### 2.4. Determination of Spermatozoa DNA Status

The SCSA technique is based on acridine orange stain, which fluoresces green when combined with double-stranded DNA, and red when combined with single-stranded DNA (denatured). The SCSA was performed according to the procedure described by Evenson et al. [19]. Fresh diluted boar semen (6 mL) was diluted in 194 mL of TNE buffer (0.15 M NaCl, 0.01 M Tris HCl, 1 mM EDTA; pH 7.4). Then 400 mL of an acid detergent solution were added. Exactly 30 s after adding the acid detergent solution (0.08 M HCl, 0.15 M NaCl, 0.1% Triton X-100; pH 1.2), 1.2 mL of staining solution (6 mg/mL of acridine orange in a buffer containing 37 mM citric acid, 126 mM Na_2_ HPO_4_, 1.1 mM disodium EDTA and 150 mM NaCl; pH 6) was added. Samples were incubated for 3 min at 37 °C and, subsequently, the sample was run through a flow cytometry. The flow cytometer (Guava Millipore, Easy Cite Mini, software Cytosoft version 4.4 beta 1; Hayward, CA, USA) with built-in software for semen quality analysis (IMV Technologies, L’Aigle, France), used a 488 nm coherent sapphire blue diode laser and photomultipliers. The DNA fragmentation index (DFI) for each semen sample was calculated as the ratio of red fluorescence to total fluorescence. 

### 2.5. Data Analyses

All data were processed by StatSoft Statistica (v12.5) software (Tulsa, OK, USA). The distribution patterns of the obtained data were revealed by descriptive statistics, while the differences in oxidative stress status, spermatozoa DNA status and total number of motile spermatozoa recorded between the control and experimental groups of animals at the selected time points were assessed by ANOVA for repeated measures. The differences were considered significant if *p* < 0.05.

## 3. Results

### 3.1. Determination of the Antioxidative System in Seminal Plasma 

The results showing the distribution of measured oxidative stress parameters in the groups of control and experimental animals are presented in Table 1. Additionally, these data show or relate to the average of five time point collections from the control and the experimental groups.

The application of ANOVA for repeated measures showed there was no statistically significant difference between the control and experimental groups in measured total antioxidant capacity (F(9, 140) = 0.616, *p* = 0.78) or in glutathione peroxidase level (F(9, 140) = 1.01, *p* = 0.43) during the entire study (Figure 1). On the other hand, a statistically significant difference was noticed in the levels of superoxide dismutase between the two groups (F(9, 140) = 4.61, *p* = 0.001), where post hoc Tukey’s HSD test showed that a difference in superoxide dismutase existed at the end of the applied treatment (Time point 4, Figure 1). 

### 3.2. Determination of Spermatozoa Motility

#### 3.2.1. Parameters of Spermatozoa Kinematics

The results showing the distribution of measured spermatozoa kinematic parameters in the groups of control and experimental animals are presented in Table 2. These data show or relate to the average of four time point collections from the control and the experimental groups.

The application of ANOVA for repeated measures showed there was no statistically significant difference between the control and experimental groups in measured curvilinear velocity (F(7, 112) = 0.99, *p* = 0.44), straight-line velocity (F(7, 112) = 1.23, *p* = 0.29) and average path velocity level (F(7, 112) = 1.32, *p* = 0.25). The results of spermatozoa kinematics did not show any statistically significant difference between groups, but did show declines in the mean values of VCL, VSL and VAP (Table 2) in the control group.

#### 3.2.2. The Total Number of Motile Spermatozoa

The application of ANOVA for repeated measures showed a statistically significant difference in the total number (%) of motile spermatozoa between the control and experimental groups (F(7, 112) = 3.12, *p* = 0.001) assessed at different time points (Figure 2). The application of post hoc Tukey’s HSD test showed that the recorded difference was notable only at the last time point (Figure 2).

### 3.3. Determination of Spermatozoa DNA Status 

The application of ANOVA for repeated measures showed a statistically significant difference in the level of spermatozoa DNA damage (%) between the control and experimental groups (F(7, 112) = 5.19, *p* = 0.001) assessed at different time points (Figure 3). The application of post hoc Tukey’s HSD test showed differences in the level of spermatozoa damage occurred at each time point, except at the beginning of the study.

## 4. Discussion

The excessive accumulation of ROS leads to oxidative stress in spermatozoa, so the limited antioxidant system in seminal plasma and the spermatozoa become overpowered [25,26]. Excessive ROS damage the structure of the spermatozoon membrane and lead to irreversible loss of mobility [27], lipid peroxidation, and DNA damage [28]. Furthermore, boar spermatozoa are highly susceptible to oxidative stress because their plasma membrane is rich in polyunsaturated fatty acids [29]. Due to the serious harmful potential of ROS, spermatozoa depend on complex defense mechanisms. 

Hence, to protect spermatozoa from oxidative stress, seminal plasma is rich in numerous antioxidant enzymes such as superoxide dismutase and glutathione peroxidase [30,31], as well as various non-enzymatic substances such as ascorbate, urate, α-tocopherol, pyruvate, glutathione, taurine, and hypotaurine. There is strong evidence indicating that superoxide dismutase and its isoenzymes are primarily involved in suppressing free superoxide anion radicals [32,33]. Endogenous antioxidant enzymes in human seminal fluid can be further enhanced from food and fluids, supplements, and pharmaceuticals [34]. Our study revealed superoxide dismutase activity differed significantly in the seminal fluid from our control and experimental groups of boars by the end of the study. This is in agreement with the results of Argenti et al. [35], where an increase in superoxide dismutase in the summer is likely caused by the Brazilian climate and the impact of heat stress on boars during that season. Moreover, Kowalowka et al. [30] showed that the seminal plasma of boars under two years of age displayed large amounts of secretory superoxide dismutase activity. Seasonal changes showed a significant effect on superoxide dismutase activity, as higher activities occur in the autumn and spring [30]. Furthermore, it has been reported that there is reduced antioxidant capacity of the boar seminal plasma during the summer period [36]. 

Barranco et al. [37] published the mean activity levels of boar seminal fluid superoxide dismutase, ranging from 1.16 ± 0.11 to 7.2 ± 0.75 IU/mL, while in our study the mean superoxide dismutase activity in this matrix was 1.55 ± 0.35 for the experimental group, and 1.60 ± 0.48 U/mL for the control group. Additionally, glutathione peroxidase remained relatively constant in the seminal fluid over time in our study, as did total antioxidant capacity. In comparison to the results of Žaja et al. [38], where the total antioxidant capacity of different breeds of boars’ seminal fluid was: Swedish Landrace—1.40; German Landrace—1.39; Large White—1.50; Pietrain—1.31 and Pig Improvement Company hybrid—1.85 mmoL/L, the average total antioxidant capacity in our study was lower, at 1.05 mmoL/L for the experimental group, and 1.08 mmoL/L for the control group. The determination of total antioxidant capacity has been demonstrated to be relevant in humans for fertility assessment, because low levels of total antioxidant capacity are associated with infertility and abnormal semen parameters [39]. This is in agreement with [38], which reported glutathione peroxidase activity several times higher compared to superoxide dismutase activity in boar seminal fluid. The benefit of organic selenium in the boar diet lies in its efficient absorption, transport, and accumulation in body reserves. In fact, dietary selenium improved the antioxidant status of the testes and semen [40]. As the levels of major natural antioxidants (vitamin E, ascorbic acid, and carotenoids) in boar semen are comparatively low [41]; this indicates that the dietary supplementation of feed mixture with selenium at 0.6 mg Se/kg significantly improved the antioxidant potential of the breeding boar semen [12]. Lasota et al. [42] found higher glutathione peroxidase activity in seminal plasma, which is opposite to the results of our study. In the same study [42], glutathione peroxidase activity increased as boars aged, but also mechanisms that control selenium metabolism were independent of boar age.

The occurrence of DNA strand breaks, associated with the fragmentation of spermatozoa DNA and spermatozoa chromatin packaging defects, can be caused by germ cell apoptosis in the testis, incomplete epididymal spermatozoa maturation, and exposure to ROS [43]. The negative impact of heat stress on spermatozoa DNA integrity is coupled with its downstream effect on early embryo development [44]. There are, however, several additional causes of spermatozoa DNA damage, including environmental stress, toxicants, pollution, infection, poor nutrition, and low antioxidant activity in the seminal plasma [45,46]. Peña et al. [47] reported reduction in DNA damage with antioxidant supplementation, similar to the results obtained in our study. The results in our study also show the protection of DNA in spermatozoa and reduction of DNA damage at three time points in the experimental group, relative to the boar control group. The same authors [47] showed that boar diets supplemented with custom-mixed antioxidants during peak wet summer effectively reduced sperm DNA damage after 42 and 84 days of treatment, respectively. Supplementation did not improve total motility spermatozoa or several other motion parameters measured by CASA for either season (in the peak wet and early dry) [47], while in our study there was improved total motility of boar spermatozoa. The combination of these two antioxidants (organic selenium and Oxynat 3D) functions to protect spermatozoa DNA. In another example, of combined use of antioxidants, selenium and Vitamin E tend to produce better results in improving boar spermatozoa motility, concentration, and/or morphology when given together than when used separately [48]. The combination of our two antioxidants, as well as the combination of other antioxidants [47,48], suggests a compound of antioxidants in a supplement formula appears to have more beneficial for boar spermatozoa. Nevertheless, we should be careful with the usage of antioxidants and their combinations, as there are studies in which the addition of various antioxidants has led to increased damage to spermatozoa DNA [47,49,50].

The results of our study also indicate an improvement in overall spermatozoa motility. Many other studies have also shown the benefits of adding antioxidants, with consequent improvements in the quality of motility and spermatozoa concentration, whether antioxidants were supplemented through boars’ food or were added to seminal fluid before storage [48,51,52,53,54]. Conversely, there are studies in which there has been no improvement in spermatozoa motility [24]. Spermatogenesis in boars lasts around 40 days, whereas the sperm transit through the epididymis takes approximately 10 days [55], the results of our study indicate an improvement in overall spermatozoa motility subsequent to the total duration of this process. In comparison to the results of Barquero et al. [56], where the spermatozoa kinematic parameters (general means) of two terminal boar sire lines (Pietrain and Duroc x Pietrain) were VCL—69.71, VSL—31.56 and VAP—41.94 µm/s, while in our study, the means obtained were VCL—64.62, VSL—47.93 and VAP—56.05 µm/s for the experimental group, and VCL—60.46, VSL—43.26 and VAP—51.06 µm/s for the control group. Actually, reduced spermatozoa motility in vitro was also reported when sodium selenite was used as a shelf-life extender [57]. Moreover, one study proved that heat stress did not affect spermatozoa motility [58], whereas Apić et al. [59] showed the strong influence of season on the ejaculate quality parameters. Namely, the values of all parameters (volume, spermatozoa concentration, the total number of spermatozoa in the ejaculate, progressive motility, as well as the number of good ejaculates) were significantly higher in cold, compared to the warm season of the year. 

## 5. Conclusions

To the best of our knowledge, this study is the first in which the ejaculate parameters were examined after the application of a combination of organic selenium and Oxynat 3D. Supplementation of boar feed with added these two antioxidant supplements improved the levels of measured superoxide dismutase activity in seminal fluid, spermatozoa DNA status, and the total number of motile spermatozoa. On the other hand, no statistically significant difference was recorded in glutathione peroxidase activity or total antioxidant capacity in seminal fluid from the animals treated with supplemented feed vs. that from the control group of boars. Furthermore, the combination of these two antioxidants, with the previously mentioned concentration of organic selenium and Oxynat 3D, 250 g/t and 500 g/t, respectively, enhanced the quality of total motility spermatozoa and DNA status in boar semen. 

Future research should be conducted in order to widen the spectrum of oxidative stress markers, and to obtain better insight into possible mechanisms of spermatozoa quality and fertility improvement in relation to the addition of dietary antioxidants.

## Data Availability

Not applicable.

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
