# Peer review of "Effect of an Antioxidant Supplement Combination on Boar Sperm"

_animals, 2022, doi:10.3390/ani12101301_

Round 1

Reviewer 1 Report

The present work addresses the effect of an Antioxidant Supplement Combination on Boar. This study is interesting and provides a very complete picture of the effect of such supplement on sperm.

However, the authors should further explain the novelty of the present work. The figures could also be improved. Some specific points about the manuscript presentation are included below.

Introduction
The introduction requries additional backgournd regarding the association beetween antioxidant capacity and DNA damage. In addition, the manuscript would be improved by the inclusion of a clear working hypothesis.
The authors should clarify why it is necessary to carry out this work because the novelty of the present study is not clear.

Materials and methods

The reviewer suggest to add a schematic presentation of the experimental design in order to improve the manuscript. 

What is the rationale for amount of antioxidants supplementation? Why was a two month regime assessed and why were the time points selected? Is there an industrial benefit? 

Additional information regarding the  Randox kits is needed.

Please add inforamation regarding the materials used in the study (i.e, company, country). 

Results

It seems that Tables 1 and 2 represents the avarage of 5 points collections of the control and the experminet groups, if so, clarify this point.

The information in the parentheses such as in lines 163-164, 166 is not clear. Please clarify it. Make sure the write the correct p values, rather than p=0.000. 

In figure 1, the control group is colored in red, while in figures 2 and 3 in blue. Is it a mistake? make sure to be consistent with the describing colors witihn the figures.

The first section of the results (i.e.,3.1. Determination of the Antioxidative System in Seminal Plasma) starts with the description of the overall effect of the antioxidant, followed by description of all 5 points through the experiment, However, section 3.2 begins with the description of 5 points.

The authors should be consistent in the data presentation.   

It is well accepted to use differnet letters when representing statistical difference, please correct the letters within figures that will represent statistical difference between groups (i.e., "a" for control and "b" for the experimental group).

The revirewer strongly suggest the authors to include a ROS examination experiment using the IMV flow cytometer, to represent the association between DNA fragmntation index within sperm and their ROS oxidation status.  

Can the author ellaborate regarding the impact of antioxidants on additional sperm parameters, such as sperm concentration, viability, progressive motility? As well as acrosome integrity, mitochondrial membrane? 

Discussion

The reviewer suggest that the author will adress the posibble association between the lenthly of the experiment and the time course of spermatogensis, as the main effect of the supplement was presented at the end of the experiment. 

Line 254: The decrese is DNA fragmentation was recorded at the first time point, but alteration in SOD was recorded at the fifth time point. In addition, no examination of ROS was conducted in the study. Please clarifiy the statment.  

Author Response

Thank you for your insightful comments and suggestions that helped us significantly improves the quality of our manuscript. We strongly believe we have managed to address each of your concerns and comments. We have addressed each of concerns as outlined below and we tried to state point-by-point the changes we have made to the manuscript. Regarding better understanding, all changes related to your comments in the text are marked in red color while with the blue color is marked change related to your and the other reviewer.

Reviewer 2 Report

In this paper, Galic and colleagues analyzed the effect of antioxidant (selenium and Oxynat 3D) supplementation, added to standard feed, on boar sperm quality. They found a higher superoxide dismutase activity as compared to the controls, as well as lower DNA fragmentation an improved motility.

The field is worthy of investigation; however, some revisions should be made to the text:

  • The abstract is little informative, as the main purpose of the study, as well as a conclusion about the significance of the obtained results are missing;
  • In line 44, the authors should specify why the use of antioxidants is still under debate;
  • In the introduction, it should be better explained the effects of ROS and oxidative stress on sperm, both in positive (capacitation, acrosome reaction) and negative (loss of membrane fluidity, DNA fragmentation) way;
  • In the section 2.2, the reference code for the used kits should be given;
  • In the section 3.2, the description of the results presented in Table 2 is completely missing;

My major concerns regard the treatments:

  • Why did the authors choose just selenium and Oxynat 3D? And why they used those specific concentrations? Why the substances were added to feed and not to water? Why they were given just in combination and not also separately?

Moreover, since the aim of the work is to study ameliorative agents for AI, why did the authors give the antioxidants to boars and not as extenders to semen? It should be surely simpler and cheaper.

Finally, I suggest a revision of English language, as throughout the text are mistakes, repetition and sometimes, sentence are difficult to follow.

Author Response

Thank you for your insightful comments and suggestions that helped us significantly improves the quality of our manuscript. We believe we have managed to address each of your concerns and comments. We have addressed each of concerns as outlined below and we tried to state point-by-point the changes we have made to the manuscript. Regarding better understanding, all changes related to your comments in the text are marked in green color while with the blue color is marked change related to your and the other reviewer.

Round 2

Reviewer 1 Report

All comments have been addressed

Reviewer 2 Report

The authors responded to all the raised issues, improving the quality of the MS. It can be published in this form in Animals.